# The Moderating Effects of Host Country Governance and Trade Openness on the Relationship between Cultural Distance and Financial Performance of Foreign Subsidiaries in Latin America

Henrique Correa da Cunha , Nursel Selver Ruzgar and Vikkram Singh *

Department of Global Management Studies, Ted Rogers School of Management, Ryerson University, 350 Victoria Street, Toronto, ON M58 2K3, Canada; henrique.cunha@ryerson.ca (H.C.d.C.); nruzgar@ryerson.ca (N.S.R.)
* Correspondence: vik.singh@ryerson.ca

**Abstract:** Cultural distance (CD) is an important driver of foreign expansion strategy at the firm level. However, its effects can be more or less significant depending on the contextual characteristics of the host country, such as the quality of formal institutions and the openness to international trade. Therefore, it is argued that strong formal institutions in the host country can effectively reduce the adverse impact of CD. Additionally, due to the more frequent interactions with foreign cultures, countries open to foreign trade can positively accommodate the effects of CD. The study tests these assumptions using data from the Orbis database and the World Bank and finds a reduction in the adverse impact of CD on the financial performance of foreign subsidiary firms with robust formal institutions in the host country. Moreover, the negative effects of CD increase with higher degrees of trade openness. Thus, the results indicate that foreign subsidiary firms operating in host countries that are more open to foreign trade will have to conform to the higher expectations from the local culture.

**Keywords:** cultural distance; formal institutions; host country governance; trade openness; Latin America; foreign subsidiary firms; financial performance

## 1. Introduction

The international business literature identifies cultural distance (CD) as an important driver of globalization strategy, for example, choice of location, degree of ownership, and entry and establishment goals (Kirkman et al. 2006; Beugelsdijk et al. 2018; Wang and Larimo 2020). However, there are concerns regarding the theoretical arguments, methodological procedures, and empirical results found in CD studies (Shenkar 2001; Dow 2017; Verbeke et al. 2017). According to Shenkar et al. (2020), to advance the knowledge of international business, there is a need to identify the conditions that allow the cultural traits to shape the internationalization outcomes. The study contributes to this debate by focusing on how formal institutions and the degree of trade openness in the host country moderate the effects of CD on the financial performance of foreign subsidiary firms.

Formal constraints are rules that human beings devise, while informal constraints include conventions and codes of behavior (North 1990). The institutions can be classified into "three pillars": regulatory, cognitive, and normative (Scott 1995) to understand how they impact firm behavior. The regulatory pillar includes laws and regulations that provide stability and order in societies. The cognitive pillar represents "world views" (i.e., how individuals in society relate to things). The normative pillar includes social values, norms, and culture. Kostova (1996) adopts this classification and defines institutional distance as the difference between two countries regarding their regulatory, cognitive, and normative institutions. According to Kostova et al. (2020, p. 470), "when companies do business across borders, they face a challenge not only to learn new ways of conducting certain functions

but also to satisfy multiple, different, and possibly conflicting, legitimacy requirements and expectations."

Culture is an essential part of informal institutions (Peng et al. 2009), with CD representing the "extent to which the shared norms [ideas, beliefs] and values in one country differ from those in another" (Drogendijk and Slangen 2006, p. 362). Comparisons between countries can involve informal (Slangen and Beugelsdijk 2010) and formal institutions, including laws, regulations, and enforcement characteristics (Estrin et al. 2009; Slangen and Beugelsdijk 2010). Formal institutions refer to governance mechanisms that govern and support business activity and human interaction (North 1990; Pejovich 1999; Correa da Cunha et al. 2022b). However, it is wrong to compare countries based on "better" or "worse" cultural traits. Instead, such observations need to consider the formal institutional environment (North 1990), as some countries have more developed and supportive formal institutions that lower the costs of operating in such environments (Cuervo-Cazurra and Genc 2011).

With the implications of CD, most studies emphasize the negative effects (Singh et al. 2017; Beugelsdijk et al. 2018). However, these effects depend on the standpoint of the observer, which might influence the effects to be more or less significant (Magnani et al. 2018; Selmer et al. 2007; Zaheer et al. 2012). The asymmetric effects of CD have been identified quantitatively by Correa da Correa da Cunha et al. (2020), showing the importance of the size and direction of CD. Moreover, the asymmetric effects of CD are likely to be associated with the different dimensions (Pizzi et al. 2021) and other important characteristics of the host country profile (Correa da Cunha 2019).

By considering the hierarchy in the institutional framework, formal institutions can be "enacted to modify, revise, or replace informal constraints" (North 1990, p. 47). By considering the direction of formal institutional distances (FID), Correa da Cunha (2019) has shown that the effects of CD tend to increase when foreign subsidiary firms operate in host countries with less developed formal institutions. Therefore, by considering the contextual characteristics of the host country, it is argued that countries with more developed and supportive formal institutions, such as strong governance mechanisms, are more effective in curbing the adverse effects of CD.

Furthermore, as some countries are more open to foreign trade, firms operating in countries with varying degrees of trade openness are affected in various ways by CD. Trade openness represents the country's orientation towards foreign trade, i.e., it measures the ratio of imports and exports compared to the size of the economy. Lee and Wen (2020) have shown that trade openness affects competition by attracting foreign companies to a country. Countries that are more open to foreign trade interact more frequently with different cultures—"the more open a country is to the trade, the more likely it is to possess culture conducive to increased social and economic interactions" (Coyne and Williamson 2009, p. 4). Thus, in countries that are more open to foreign trade, the more frequent and intense interactions with different cultures will lower the effects of CD; whereas, for those with lower trade openness, the impact of CD is higher. We test these assumptions using secondary data from the Orbis database with over 1400 foreign subsidiaries from developed countries and emerging markets operating in the ten largest economies in Latin America.

This study advances the knowledge of the conditions under which the effects of CD can be more or less significant by highlighting the different moderating effects of formal institutions and the degree of trade openness in the host country. Although cultural values are known to remain reasonably stable over time (Hofstede 1980), formal institutions can change faster, causing the effects of CD to increase when formal institutions in the host country deteriorate or decrease when governments work towards implementing strong formal institutions. By highlighting how the contextual characteristics in the host country interact with CD, this study attempts to reconcile the conflicting views (e.g., symmetric and asymmetric) on how CD affects the outcomes of internationalization. Moreover, the study shows that focusing exclusively on the direct effects provided by CD can be problematic as it fails to account for other important contextual factors of the study. In that sense, as

noted by Whetten (1989, p. 492). "temporal and contextual factors set the boundaries of generalizability, and as such constitute the range of the theory."

Results regarding the negative effects of CD on the financial performance of foreign subsidiaries in Latin America are in line with previous research (Singh et al. 2017; Beugelsdijk et al. 2018). However, findings reveal that the adverse effects of CD are moderated negatively by formal institutions in the host country. Therefore, strong formal institutions provide an effective mechanism to curb the adverse effects of CD on the financial performance of foreign subsidiary firms. Furthermore, contrary to the hypothesis, we find that the negative impacts of CD increase in countries that are more open to foreign trade. The increased negative effects of CD in countries that are more open to foreign trade are likely to be associated with the higher requirements on foreign subsidiary firms to conform to the expectations and requirements of the local culture.

Following the introduction (Section 1), the remaining parts of this study are organized into five major sections. Section 2 presents the theoretical background and hypotheses. The data and method are presented in Section 3. Section 4 discusses the results, and Section 5 provides the conclusions, limitations, and directions for future research.

## 2. Literature Review, Theoretical Background and Hypothesis

Institutions can be classified according to formality (North 1990) into three pillars (Scott 1995). According to Peng et al. (2009), formal institutions incorporate the regulatory framework and include formal rules, regulations, laws, and enforcement mechanisms and characteristics. Informal institutions encompass the normative and cognitive pillars and include norms, culture, and ethics. Moreover, culture is at the core of informal institutions (Peng et al. 2009).

Formal institutions refer to "the rules of the game in a society or, more formally, are the humanly devised constraints that shape human interaction. They structure incentives in human exchange, whether political, social, or economic" (North 1990, p. 3). A hierarchy exists within the different levels of formality in the institutional framework, such that formal institutions can be "enacted to modify, revise, or replace informal constraints" (North 1990, p. 47). Therefore, it is crucial to consider the interactions with formal constraints when investigating the effects of informal institutions.

### 2.1. Cultural Distance and Performance

Studies have shown the relationship between cultural characteristics and economic development (Hofstede et al. 2010; Correa da Cunha et al. 2022a). Moreover, studies have shown that cultural traits can contribute to agency costs (Orlova 2020). Additionally, foreign firms are at a disadvantage (Zaheer 1995) due to the cultural distance between home and host countries (Kogut and Singh 1988). Cultural distance represents the "extent to which the shared norms [ideas, beliefs] and values in one country differ from those in another" (Drogendijk and Slangen 2006, p. 362). Most studies emphasize their negative effects (Beugelsdijk et al. 2018) because the distance represents the liability of foreignness (Zaheer 1995) as it creates friction (Shenkar et al. 2008; Shenkar 2012) and increases the costs of doing business abroad (Cuervo-Cazurra and Genc 2011). We test the following hypothesis to assess the effects of CD on the financial performance of foreign subsidiary firms:

**H1.** *CD lowers the financial performance of foreign subsidiary firms in Latin America.*

### 2.2. The Moderating Effects of Host Country Formal Institutions

Formal institutions are strong "if they support the voluntary exchange underpinning an effective market mechanism" and weak "if they fail to ensure effective markets or even undermine markets" (Meyer et al. 2009, p. 63). Studies show that more developed formal institutions in the host country provide better conditions for firms to operate as it lowers transaction costs and increases financial performance (Cuervo-Cazurra and Genc 2011; Maseland 2013; Zaheer et al. 2012; Hernández and Nieto 2015; Konara and Shirodkar 2018).

According to North (1990), formal and informal institutions can interact in a complementary or suppressive manner. Institutional complementarity refers to a condition in which "the presence (or efficiency) of one [institution] increases the returns from (or efficiency of) the other" (Hall and Soskice 2001, p. 17). Alternatively, the suppressive relationship refers to a condition in which the formal rules may be "enacted to modify, revise, or replace informal constraints" (North 1990, p. 47). According to Pejovich (1999, p. 170), "formal institutions suppress, but fail to change informal institutions." Furthermore, "formal rules change, but the informal constraints do not" (North 1990, p. 91). Thus, we argue that, due to the higher hierarchy and priority of formal institutions, the strength of formal institutions and governance characteristics in the host country conditions the effects of CD. When governance is stronger, it lowers the effects of CD and vice-versa. To test these assumptions, we propose the following hypothesis:

**H2.** *The stronger the governance characteristics in the host country, the lower the negative effects of CD on the financial performance of foreign subsidiaries.*

### 2.3. The Moderating Effects of Host Country Trade Openness

Trade openness represents the country's orientation towards foreign trade. It measures the ratio of imports and exports compared to the size of the economy. Reyes et al. (2019) show that trade openness affects inward foreign direct investment in Latin America and, thus, creates incentives for foreign firms to invest in the region. Countries that are more open to foreign trade interact more frequently with other cultures. For instance, Lee and Wen (2020, p. 140) identified that "the government-imposed trade liberalization significantly increases the entry threat from foreign competitors, offering a specific setting in which to capture the dynamic nature of changing competition over time." Coyne and Williamson (2009, p. 4) state that "the more open a country is to the trade, the more likely it is to possess culture conducive to increased social and economic interactions."

Thus, we argue that firms operating in countries with different degrees of trade openness face the varying impact of CD. The more frequent and intense interaction with foreign cultures might explain why the effect of CD in more open host countries is lesser than those in closed economies. Hypothesis H3 tests this argument:

**H3.** *The higher the degree of trade openness, the lower the negative effects of CD on the financial performance of foreign subsidiaries.*

### 3. Methodology

This study uses a quantitative approach using the econometric technique of panel data to test the hypotheses. Due to fast-paced and highly volatile conditions in Latin America, panel data can check the consistency in patterns and relationships for the variables included in this study. The data on foreign subsidiaries is from the Orbis database. However, not all firms provide complete data or participate in the survey every year, leading to missing data and unbalanced data panels. The final sample for this study is selected considering the tradeoff between a larger number of individual subsidiary firms over a shorter period and a larger number of individual firms over an extended period. The final sample comprises foreign subsidiaries operating in the ten largest economies in Latin America: Argentina, Brazil, Colombia, Chile, Ecuador, Mexico, Panama, Peru, Uruguay, and Venezuela, over three consecutive years from 2013 to 2015, totaling over 4200 firm-year observations.

Due to this vast diversity, Latin America provides a relevant context for this study as "societal, cultural, and economic characteristics that make the region an ideal 'natural laboratory' to build and test management theories" (Aguinis et al. 2020, p. 615). Observing the same individual firms over three consecutive years provides more robust estimates because of the rapidly evolving domestic economic and political conditions in Latin America. Furthermore, bootstrap analysis improves the stability of the parameter estimates given the reduced number of individual subsidiary firms (Efron and Gong 1983). When comparing

bootstrap estimates with the original dataset, the significance of the parameter estimates improves in the bootstrap sample. Also, the results in both samples are consistent for all empirical tests.

### 3.1. Dependent Variable

We use profit margins, a widely cited measure in literature, to measure the financial performance of the subsidiaries (Venkatraman and Ramanujam 1986; Hitt et al. 1997; Correa da Cunha et al. 2020). Furthermore, the profit margin is less susceptible to different asset valuations that result from the time of investment or depreciation (Michael Geringer et al. 1989, Contractor et al. 2003). When comparing firms in different industries, profit margin provides a more equitable alternative to measure firm performance as firms in various sectors use assets differently. In turbulent contexts, such as in emerging markets, sustaining the company's profit margins become even more challenging and reflects management's effectiveness at investing in projects that add value (Chopra and Mier 2017). The profit margin data is obtained from the Obis database.

### 3.2. Independent Variables

(a) Cultural Distance (CD): Calculated using the Kogut and Singh (KS) (Kogut and Singh 1988) composite index using the four original dimensions of Hofstede (1980): individualism vs. collectivism, power distance, masculinity vs. femininity, and uncertainty avoidance.

(b) Host Country Governance (Host Country Formal Institutions): In line with previous research, the quality of formal institutions in the host country is measured using the World Governance Indicators (WGI) from the World Bank developed by Kaufmann et al. (2009). The WGI is closely related to the normative and regulatory pillars and is extensively used in literature to assess the strength of formal institutions (Stein and Daude 2001; Globerman and Shapiro 2003; Gani 2007; Wernick et al. 2009; Mengistu and Adhikary 2011) The WGI includes six variables: voice and accountability (VOICE), political stability and absence of violence (POL), government effectiveness (GOV), regulatory quality (REG), rule of law (RULE) and control of corruption (CC)—they represent "the traditions and institutions by which authority in a country is exercised" (Kaufmann et al. 2011, p. 4). Due to the high correlation among the six WGI variables, the strength of host country governance is computed as a composite index calculated as the arithmetic means of the six WGI variables.

(c) Trade Openness Index: Following previous studies, the trade openness index measures the sum of imports and exports as a percentage of total GDP (Kolstad and Wiig 2012; Reyes et al. 2019). The index data is collected from the World Bank.

### 3.3. Control Variables

Several controls known to affect the financial performance of foreign subsidiary firms are included at the subsidiary, home, and host country-level: size of the economy, trade openness, country-level governance, industry sector (i.e., industrial vs. service firms), industry sector annual growth, subsidiary annual sales growth, subsidiary size, and subsidiary market share (Capon et al. 1990; Dikova 2009; Cuervo-Cazurra and Genc 2011; Hernández and Nieto 2015; Konara and Shirodkar 2018). Figure 1 illustrates the general framework of the study.

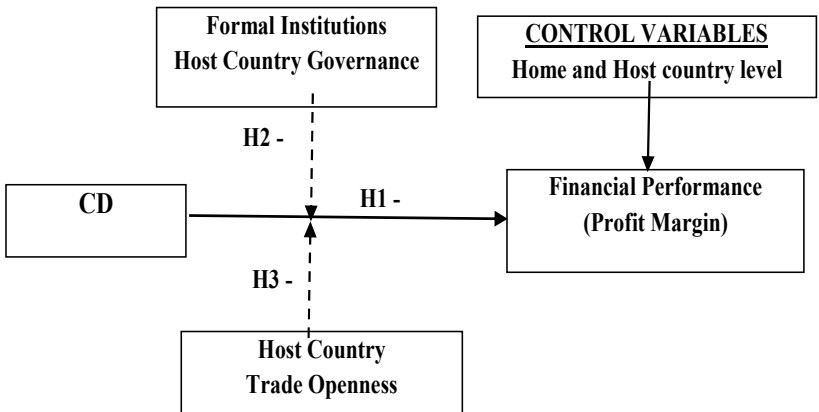

**Figure 1.** The moderating effects of formal institutions and trade openness on the relationship between CD and financial performance of foreign subsidiaries.

*3.4. The Moderation Tests*

The moderation tests are performed by analyzing how changes in the moderator variable affect the coefficients. We follow the procedure described by Hayes (2013) to test hypotheses H2 and H3—it involves evaluating the change in the slope of the regression line that results from adding and subtracting one standard deviation from the moderator variable.

**4. Results**

We use the random effects (REs) estimation method to "control for unobserved heterogeneity because some of the independent variables are time-invariant" (Salomon and Wu (2012, p. 359). Therefore, the REs allows verifying the effects of CD, which remain constant over time. Additionally, White's test (Andrews 1991; MacKinnon and White 1985; White 1980) indicates no heteroskedasticity issues. Moreover, Lu and White (2014, p. 178) indicate that "a now common exercise in empirical studies is a 'robustness check,' where the researcher examines how certain 'core' regression coefficient estimates behave when the regression specification is modified in some way, typically by adding or removing regressors." The consistency of the results is verified as the size and signal for the coefficients and the characteristics of the models remain unchanged when new variables are introduced during the moderation tests. Values lower than 2 for the variance inflation factor (VIF) test for all variables show no multicollinearity issues (Rogerson 2001). Table 1 shows the test results for hypotheses H1 and H2.

The negative and statistically significant effect of cultural distance (CD_KS) for all empirical arrangements show that the cultural distance negatively affects the financial performance of firms in Latin America (Table 1), supporting hypothesis H1. Furthermore, results reveal that host country governance has a positive and significant direct effect on the financial performance of foreign subsidiary firms. Moreover, the results support hypothesis H2 as the strength of governance in the host country negatively moderates the effects of CD. The decrease in the coefficient of the main relation (between CD_KS and financial performance) verifies the negative moderation. It changes from statistically significant $-1.033$ ($p$-value < 0.01) at low governance levels ($-1$ std. deviation) to $-0.806$ ($p$-value < 0.01) at moderate levels and the effects become even lower ($-0.578$ and $p$-value < 0.05) when host country governance becomes stronger ($+1$ std. deviation). Figure 2 shows the change in the slope of the regression line that represents the effects of CD on the profit margin of foreign subsidiaries firms when the quality of governance in the host country improves (solid line, $+1$ std. deviation) and worsens (dotted line, $-1$ std. deviation).

**Table 1.** The effect of CD and the moderating effects of host country governance on the relation between CD and performance.

| | Hypothesis H1 | Hypothesis H2 | | |
|---|---|---|---|---|
| | **CD** | **Governance Low (−1 std. Deviation)** | **Governance Intermediate** | **Governance High (+1 std. Deviation)** |
| Const | −88.191 *** | −87.036 *** | −87.945 *** | −88.855 *** |
| | (7.301) | (7.293) | (7.301) | (7.311) |
| | [0.000] | [0.000] | [0.000] | [0.000] |
| Industry or Service (dummy) | −1.444 *** | −1.518 *** | −1.518 *** | −1.518 *** |
| | (0.382) | (0.383) | (0.383) | (0.383) |
| | [0.000] | [0.000] | [0.000] | [0.000] |
| Total Assets | 0.000 *** | 0.000 *** | 0.000 *** | 0.000 *** |
| | (0.000) | (0.000) | (0.000) | (0.000) |
| | [0.000] | [0.000] | [0.000] | [0.000] |
| Sales Revenues | −0.000 | −0.000 | −0.000 | −0.000 |
| | (0.000) | (0.000) | (0.000) | (0.000) |
| | [0.782] | [0.844] | [0.844] | [0.844] |
| Market Share | −27,525.038 *** | −27,583.179 *** | −27,583.179 *** | −27,583.179 *** |
| | (6637.171) | (6636.949) | (6636.949) | (6636.949) |
| | [0.000] | [0.000] | [0.000] | [0.000] |
| Subsidiary Annual Sales Growth | −0.003 *** | −0.003 *** | −0.003 *** | −0.003 *** |
| | (0.001) | (0.001) | (0.001) | (0.001) |
| | [0.001] | [0.001] | [0.001] | [0.001] |
| Industry Annual Growth | 0.023 | 0.019 | 0.019 | 0.019 |
| | (0.048) | (0.048) | (0.048) | (0.048) |
| | [0.632] | [0.694] | [0.694] | [0.694] |
| Home Country | | | | |
| Home Country GDP | 0.479 | 0.527 | 0.527 | 0.527 |
| | (0.321) | (0.322) | (0.322) | (0.322) |
| | [0.136] | [0.102] | [0.102] | [0.102] |
| Home Country Trade Openness | 0.004 | 0.004 | 0.004 | 0.004 |
| | (0.003) | (0.003) | (0.003) | (0.003) |
| | [0.298] | [0.225] | [0.225] | [0.225] |
| Home Country Governance | 1.621 *** | 1.569 *** | 1.569 *** | 1.569 *** |
| | (0.344) | (0.344) | (0.344) | (0.344) |
| | [0.000] | [0.000] | [0.000] | [0.000] |
| Host Country | | | | |
| Host Country GDP | 7.144 *** | 7.092 *** | 7.092 *** | 7.092 *** |
| | (0.546) | (0.546) | (0.546) | (0.546) |
| | [0.000] | [0.000] | [0.000] | [0.000] |
| Host Country Trade Openness | 0.224 *** | 0.223 *** | 0.223 *** | 0.223 *** |
| | (0.014) | (0.014) | (0.014) | (0.014) |
| | [0.000] | [0.000] | [0.000] | [0.000] |

**Table 1.** *Cont.*

| | Hypothesis H1 | | Hypothesis H2 | |
| --- | --- | --- | --- | --- |
| | **CD** | **Governance Low (−1 std. Deviation)** | **Governance Intermediate** | **Governance High (+1 std. Deviation)** |
| Host Country Governance | 4.583 *** | | 6.434 *** | |
| | (0.452) | | (0.714) | |
| | [0.000] | | [0.000] | |
| CD_KS (main effect under investigation) | −0.664 *** | −1.033 *** | −0.806 *** | −0.578 ** |
| | (0.245) | (0.269) | (0.249) | (0.247) |
| | [0.007] | [0.000] | [0.001] | [0.019] |
| Host Country Governance Low | | 6.434 *** | | |
| | | (0.714) | | |
| | | [0.000] | | |
| CD_KS * Host Country Governance Low | | −1.611 *** | | |
| | | (0.480) | | |
| | | [0.001] | | |
| CD_KS * Host Country Governance Intermediate | | | −1.611 *** | |
| | | | (0.480) | |
| | | | [0.001] | |
| Host Country Governance High | | | | 6.434 *** |
| | | | | (0.714) |
| | | | | [0.000] |
| CD_KS * Host Country Governance High | | | | −1.611 *** |
| | | | | (0.480) |
| | | | | [0.001] |
| Number of observations | 47,714 | 47,714 | 47,714 | 47,714 |
| Adj. R2 | 0.031 | 0.033 | 0.033 | 0.033 |
| P-value(F) | 0.000 | 0.000 | 0.000 | 0.000 |

Notes: random effects (GLS) estimates; dependent variable: profit margin; ** $p < 0.05$; *** $p < 0.01$. Standard errors in parentheses and *p*-values in brackets. CD_KS refers to cultural distance as per Kogut and Singhs' index.

Table 2 displays the changes in the average of the six WGI variables in the countries included in this study, ranging from 1996 (the first year the indicators were available) to 2020 in intervals of five years (when available). In countries such as Argentina, Brazil, Chile, Ecuador, Mexico, Panama and Venezuela, the deterioration in the quality of formal institutions is associated with an increase in the negative effects of CD experienced by foreign subsidiary firms operating in the region. On the other hand, a few exceptions, such as Colombia, Peru, and Uruguay, show some improvements in the quality of formal institutions. In these three countries, the effects of CD in 2020 are expected to be lower compared to 1996.

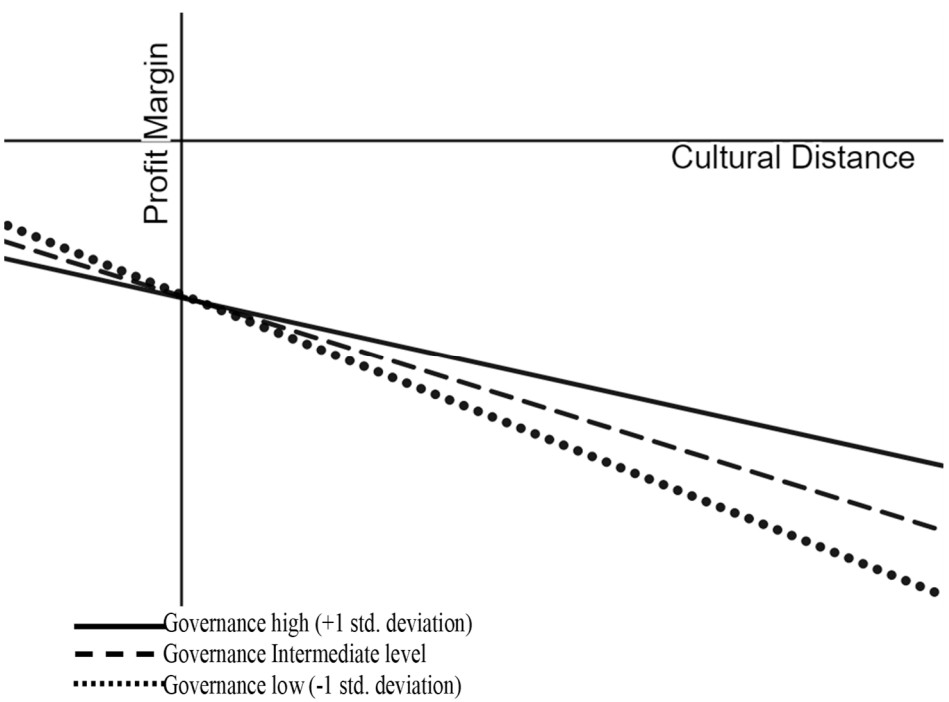

**Figure 2.** The moderating effects of host country governance on the CD-performance relationship.

**Table 2.** Average for the six WGI variables ranging from 1996 to 2020.

|              | 1996  | 2000  | 2005  | 2010  | 2015  | 2020  |
|--------------|-------|-------|-------|-------|-------|-------|
| Argentina    | 0.19  | 0.08  | −0.23 | −0.27 | −0.31 | −0.12 |
| Brazil       | −0.01 | 0.13  | −0.07 | 0.13  | −0.13 | −0.21 |
| Chile        | 1.14  | 1.14  | 1.25  | 1.22  | 1.08  | 0.89  |
| Colombia     | −0.66 | −0.59 | −0.55 | −0.36 | −0.20 | −0.14 |
| Ecuador      | −0.45 | −0.61 | −0.73 | −0.78 | −0.61 | −0.46 |
| Mexico       | −0.31 | −0.01 | −0.10 | −0.17 | −0.25 | −0.41 |
| Panama       | 0.13  | 0.21  | 0.02  | 0.09  | 0.19  | 0.08  |
| Peru         | −0.36 | −0.34 | −0.43 | −0.24 | −0.16 | −0.10 |
| Uruguay      | 0.76  | 0.76  | 0.68  | 0.84  | 0.86  | 0.97  |
| Venezuela, RB| −0.52 | −0.60 | −1.03 | −1.29 | −1.43 | −1.82 |

Notes: World Governance Indicators (WGI) range from −2.5 (weak) to +2.5 (strong).

Table 3 presents the test results for hypothesis H3, which relates to the moderating effects of trade openness on the relationship between CD and the financial performance of foreign subsidiary firms.

**Table 3.** The moderating effects of host country trade openness on the relation between CD and performance.

|  | Hypothesis H3 | | |
|---|---|---|---|
|  | **Trade Openness Low (−1 std. Deviation)** | **Trade Openness Intermediate** | **Trade Openness High (+1 std. Deviation)** |
| Const | −81.870 *** | −84.080 *** | −86.290 *** |
|  | (7.294) | (7.388) | (7.491) |
|  | [0.000] | [0.000] | [0.000] |
| Industry or Service (dummy) | −1.437 *** | −1.437 *** | −1.437 *** |
|  | (0.382) | (0.382) | (0.382) |
|  | [0.000] | [0.000] | [0.000] |
| Total Assets | 0.000 *** | 0.000 *** | 0.000 *** |
|  | (0.000) | (0.000) | (0.000) |
|  | [0.000] | [0.000] | [0.000] |
| Sales Revenues | −0.000 | −0.000 | −0.000 |
|  | (0.000) | (0.000) | (0.000) |
|  | [0.701] | [0.701] | [0.701] |
| Market Share | −22,801.866 *** | −22,801.866 *** | −22,801.866 *** |
|  | (6763.173) | (6763.173) | (6763.173) |
|  | [0.001] | [0.001] | [0.001] |
| Subsidiary Annual Sales Growth | −0.003 *** | −0.003 *** | −0.003 *** |
|  | (0.001) | (0.001) | (0.001) |
|  | [0.001] | [0.001] | [0.001] |
| Industry Annual Growth | 0.029 | 0.029 | 0.029 |
|  | (0.048) | (0.048) | (0.048) |
|  | [0.549] | [0.549] | [0.549] |
| Home Country |  |  |  |
| Home Country GDP | 0.473 | 0.473 | 0.473 |
|  | (0.321) | (0.321) | (0.321) |
|  | [0.141] | [0.141] | [0.141] |
| Home Country Trade Openness | 0.004 | 0.004 | 0.004 |
|  | (0.003) | (0.003) | (0.003) |
|  | [0.267] | [0.267] | [0.267] |
| Home Country Governance | 1.642 *** | 1.642 *** | 1.642 *** |
|  | (0.344) | (0.344) | (0.344) |
|  | [0.000] | [0.000] | [0.000] |
| Host Country |  |  |  |
| Host Country GDP | 7.000 *** | 7.000 *** | 7.000 *** |
|  | (0.548) | (0.548) | (0.548) |
|  | [0.000] | [0.000] | [0.000] |
| Host Country Trade Openness |  | 0.167 *** |  |
|  |  | (0.021) |  |
|  |  | [0.000] |  |

**Table 3.** *Cont.*

| | Hypothesis H3 | | |
| --- | --- | --- | --- |
| | **Trade Openness Low (−1 std. Deviation)** | **Trade Openness Intermediate** | **Trade Openness High (+1 std. Deviation)** |
| Host Country Governance | 4.734 *** | 4.734 *** | 4.734 *** |
| | (0.454) | (0.454) | (0.454) |
| | [0.000] | [0.000] | [0.000] |
| CD_KS (main effect under investigation) | −1.738 *** | −2.173 *** | −2.609 *** |
| | (0.384) | (0.482) | (0.589) |
| | [0.000] | [0.000] | [0.000] |
| Host Country Trade Openness Low | 0.167 *** | | |
| | (0.021) | | |
| | [0.000] | | |
| CD_KS * Host Country Trade Openness Low | 0.033 *** | | |
| | (0.009) | | |
| | [0.000] | | |
| CD_KS * Host Country Trade Openness Intermediate | | 0.033 *** | |
| | | (0.009) | |
| | | [0.000] | |
| Host Country Trade Openness High | | | 0.167 *** |
| | | | (0.021) |
| | | | [0.000] |
| CD_KS * Host Country Trade Openness High | | | 0.033 *** |
| | | | (0.009) |
| | | | [0.000] |
| Number of observations | 47,714 | 47,714 | 47,714 |
| Adj. R2 | 0.033 | 0.033 | 0.033 |
| P-value(F) | 0.000 | 0.000 | 0.000 |

Notes: random effects (GLS) estimates; dependent variable: profit margin; *** $p < 0.01$. Standard errors in parentheses and *p*-values in brackets. CD_KS refers to cultural distance as per Kogut and Singhs' index.

Concerning hypothesis H3, the test results in Table 3 reveal that contrary to expectations, trade openness positively moderates the effects of CD on the financial performance of foreign subsidiaries in Latin America. This can be verified by the increase in the coefficient of the cultural distance (CD_KS) variable, which changes from −1.738 (*p*-value < 0.01) at low degrees of trade openness (−1 std. deviation) to −2.173 (*p*-value < 0.01) at intermediate values and −2.609 (*p*-value < 0.01) at high degrees of trade openness (+1 std. deviation). Figure 3 displays the positive moderating effects of trade openness on the relationship between CD and the financial performance of foreign subsidiary firms in Latin America. The change in the slope of the regression shows that the negative effects of CD on the financial performance of foreign subsidiaries increase when trade openness in the host country increase by 1 std. deviation (solid line) and decreases as the host country becomes less open to foreign trade (dotted line, −1 std. deviation).

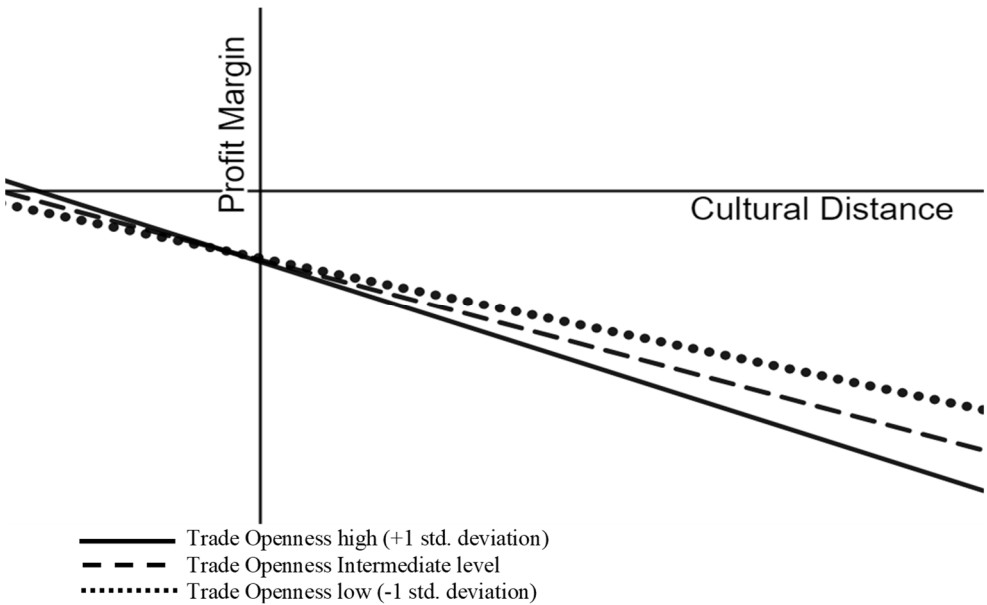

**Figure 3.** The moderating effects of trade openness on the CD-performance relationship.

These results provide a more nuanced view of the conditions under which the effects of CD can be higher or lower. By looking at the moderating effects of trade openness, these findings complement the literature regarding the asymmetric effects of CD. Magnani et al. (2018) show that exports from Brazil, a relatively closed economy with a trade openness index (exports + imports as % of GDP) lower than 30%, face more significant challenges associated with CD when doing business in Italy (a much more open economy with a trade openness index greater than 60%) than Italians adjusting to the Brazilin culture. Similarly, Selmer et al. (2007) verified that Germans find it easier to adapt to the local culture in the U.S. (trade openness index of approximately 25%), in contrast to American expatriates in Germany, which is a much more open economy to international trade with a trade openness index of over 90%. The moderating effects show that the different and asymmetric effects of CD experienced by the different groups of expatriates might result from the degree of trade openness in the host country.

Table 4 shows the changes in the trade openness index in the countries over forty years, starting in 1980 in intervals of five years. Trade openness increases in most countries in the region—the average trade openness index for Latin America and the Caribbean in 2020 is 46.6, 50% higher than 30.3 in 1980. Therefore, according to our findings, as countries in the region became more open to foreign trade over the past 4 decades, the negative effects of CD experienced by foreign subsidiary firms operating in the region might have increased during the period.

**Table 4.** Trade openness index since 1980 (in intervals of 5 years) for the countries in this study and the average in Latin America and the Caribbean.

|  | 1980 | 1985 | 1990 | 1995 | 2000 | 2005 | 2010 | 2015 | 2020 |
|---|---|---|---|---|---|---|---|---|---|
| Argentina | 11.5 | 18.0 | 15.0 | 19.8 | 22.6 | 40.6 | 35.0 | 22.5 | 30.1 |
| Brazil | 20.2 | 20.4 | 15.2 | 17.0 | 22.6 | 27.1 | 22.8 | 27.0 | 32.4 |
| Chile | 48.1 | 50.6 | 61.7 | 55.0 | 59.3 | 71.6 | 69.1 | 59.0 | 57.8 |
| Colombia | 31.8 | 26.3 | 34.8 | 35.5 | 32.7 | 37.4 | 34.3 | 38.4 | 33.7 |
| Ecuador | 35.0 | 35.7 | 44.6 | 45.9 | 59.5 | 56.1 | 60.3 | 45.2 | 43.3 |
| Mexico | 22.4 | 24.3 | 38.5 | 46.3 | 52.4 | 53.9 | 60.8 | 71.1 | 78.2 |
| Panama | 137.5 | 94.1 | 121.8 | 146.9 | 134.0 | 135.7 | 148.3 | 99.9 | 74.1 |
| Peru | 47.6 | 44.9 | 29.5 | 30.9 | 35.5 | 47.4 | 51.7 | 45.2 | 43.4 |
| Uruguay | 35.7 | 47.9 | 41.6 | 38.1 | 36.7 | 58.9 | 51.7 | 45.3 | 46.4 |
| Venezuela, RB | 57.4 | 39.4 | 57.7 | 47.3 | 47.9 | 60.1 | 46.1 |  |  |
| Latin America & Caribbean | 30.3 | 29.6 | 33.1 | 35.2 | 39.2 | 45.7 | 44.0 | 44.5 | 46.6 |

Notes: Trade openness index measures exports + imports as a percentage of GDP.

## 5. Discussion

We contribute to the knowledge of cultural distance by showing that its effects can be more or less significant depending on the characteristics of the host country. In line with previous research (Beugelsdijk et al. 2018), our findings reveal that CD represents a liability of foreignness (Zaheer 1995) to foreign subsidiary firms operating in Latin America as it affects the financial performance of these firms negatively. Furthermore, studies show that the effects of CD might be asymmetric (Shenkar 2001; Selmer et al. 2007; Zaheer et al. 2012; Magnani et al. 2018) as the same distance can have significantly different effects depending on the standpoint of the observer. This study contributes to this debate by providing a more nuanced view of the conditions under which the impact of CD can become higher or lower.

First, by analyzing how formal and informal institutions interact (North 1990; Pejovich 1999; Hall and Soskice 2001; Correa da Cunha 2019), results reveal that the effects of CD are suppressed by the strength of formal institutions in the host country. In that sense, findings show that formal institutions prioritize more than informal ones (e.g., culture). Therefore, the adverse effects of CD tend to be lower in host countries with more supportive formal institutions and stronger governance mechanisms. These results can help foreign firms understand what can be expected regarding the effects of CD by considering the characteristics of the host country and the tradeoff between entering a more culturally distant host country versus a more developed country in terms of formal institutions. Moreover, countries can become more competitive in attracting foreign investment when they provide more supportive governance structures that reduce the negative effects of cultural differences, which increases the profitability of foreign subsidiary firms.

Second, contrary to expectation, although trade openness in the host country positively affects the financial performance of foreign subsidiary firms, it positively moderates (i.e., increases) the adverse effects of CD. Thus, a higher trade openness index of a host country leads to an increase in the negative effect of CD on the financial performance of foreign subsidiary firms. These results suggest that the more open the economy to trade (i.e., imports and exports), the higher the pressure for foreign subsidiary firms to conform to the demands of the local culture, i.e., the more open the economy becomes to foreign trade, the higher the competition and the bargaining power of customers. There are more options to replace foreign subsidiary firms that fail to accommodate the needs and requirements of the local culture. Thus, countries that are more open to foreign trade might have more frequent and sometimes more diversified experiences doing business with foreign

cultures. However, it does not necessarily translate to a more tolerant environment when accommodating the cultural differences between the foreign subsidiary firm and the local environment. Host countries that are more open to foreign trade have higher competition from foreign firms (Lee and Wen 2020). Therefore, foreign subsidiary firms will face higher demands from local stakeholders to conform to the expectations of the local culture, which explains the significantly higher negative effects of CD.

This study has several relevant implications for policymakers and practitioners. These insights can assist governments to improve the attractiveness of countries in the region by implementing more supportive formal institutions. It has been shown that strong formal institutions offer positive and significant business advantages, but they can also moderate the adverse effects of CD. Thus, by improving the governance in countries in Latin America, policy formulation in this area can improve the conditions for foreign subsidiary firms to operate in the region. Furthermore, as countries in Latin America become more open to foreign trade, foreign subsidiary firms must increase their awareness regarding the adverse effects of the CD. As a result, foreign subsidiaries operating in the region can identify alternatives to adjust more favorably to the local culture.

## 6. Conclusions

This study shows that the effects of CD are highly dependent on other contextual characteristics of the host country. Investigating how formal institutions and trade openness in the host country moderates the impact of CD provides a more nuanced view of the conditions under which the effects can be higher or lower. It can be concluded that although CD between countries remains relatively stable over time, changes in formal institutions and policies that affect foreign trade can modify the way CD affects the financial performance of foreign subsidiary firms in a much shorter period. Being aware of such implications can help companies decide how and where to invest in increasing overall firm performance. Firms can evaluate the tradeoff between entering a host country that is more distant in terms of CD but has more developed formal institutions than a similar country in terms of national culture with less supportive formal institutions. Countries can develop better governance and policies that provide better conditions for foreign subsidiary firms to operate.

This study has limitations, which provide great opportunities for future research. Future research could strengthen our findings and contributions by adopting a more nuanced approach, such as the one proposed by Ionascu et al. (2004) to investigate how contextual characteristics of the host country affect the relationship between cognitive, normative, and regulatory distances. Moreover, they can focus on different contexts and specific measures of CD to account not only for the size but also for the direction of the distance. This can help advance our understanding of how different contextual elements in the host country affect the financial performance of foreign subsidiary firms.

**Author Contributions:** Conceptualization, H.C.d.C. and V.S.; methodology, H.C.d.C., V.S. and N.S.R.; software, H.C.d.C. and N.S.R.; validation, H.C.d.C., V.S., and N.S.R.; formal analysis, H.C.d.C., V.S., and N.S.R.; investigation, H.C.d.C. and V.S.; data curation, H.C.d.C. and N.S.R.; writing—original draft preparation, H.C.d.C.; writing—review and editing, H.C.d.C., V.S., and N.S.R. All authors have read and agreed to the published version of the manuscript.

**Funding:** This research received no external funding.

**Institutional Review Board Statement:** Not applicable.

**Informed Consent Statement:** Not applicable.

**Data Availability Statement:** This research utilized secondary data from the sources cited in the paper.

**Acknowledgments:** The authors are grateful for the feedback received during the paper development workshop at AIB-Canada Conference in 2021.

**Conflicts of Interest:** The authors declare no conflict of interest.

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
