# Peer review of "The Moderating Effects of Host Country Governance and Trade Openness on the Relationship between Cultural Distance and Financial Performance of Foreign Subsidiaries in Latin America"

_ijfs, doi:10.3390/ijfs10020026_

Round 1

Reviewer 1 Report

The study explores an interesting topic for readers and provides a good basis for further research. The paper shows that it is a deep and dedicated work, so I can only make minimal improvements. The structure of the paper is appropriate, and it includes the parts that are expected in an academic paper. The literature review draws on an adequate number of references. The hypotheses are well-founded, the methodological background and the results are clear. What I might suggest for reflection is Table 2, which contains a very large amount of data in relative terms. If it is possible, I would suggest splitting it up, as this would make the data contained in it easier to understand. I consider Figures 2 and 3 to be both too large and unnecessary. Having considered these suggestions, I recommend that the article be adopted.

Reviewer 2 Report

The manuscript, "The Moderating Effects of Host Country Governance and Trade Openness on the Relationship between Cultural Distance and Financial Performance of Foreign Subsidiaries in Latin America" is a well-written piece of research and provides valuable information about cultural distance and foreign expansion strategy at the firm level. As a reviewer, I suggest the following improvements:    

1. English language needs moderate improvements, like:

The following phrases appear with and without a hyphen:

  • ‘country-level’ / ‘country level’ 1 time with a hyphen 2 times without
  • ‘Random-effects’ / ‘Random effects’ 1 time with a hyphen 1 time without

Line 109: "2.1. Culture Distance and Performance" should be "cultural distance"

Line 171: provide instead of provides

Line 185: "become, reflect" instead of "becomes, reflect" respectively.

2. Abstract section showcases adequate information about the paper. In line 14, the use of "We" being the first person pronoun should not be used in scientific literature. Please amend accordingly. In the end, a short policy suggestion may also be added.

3. Line 26, Introduction section: Clarification/citation required about, "International business literature" The references provided at the end of the sentence (Kirkman et al., 2006; Beugelsdijk et al., 2018) are old enough. Please add up-to-date/ latest references. Would be better if references are furbished like: "International business literature (References) identifies........."

4. Line 33: "We attempt to contribute to this debate by focusing on the interaction of formal institutions and how that moderates CD." 

Please avoid using first-person pronouns

The methodology section is nicely presented, results are clearly explained, the discussion and conclusion are in line with the findings of the study, and finally, advice for future research is also reasonable. 

My best wishes for the authors.
